# ON THE ADVERSARIAL VULNERABILITY OF LABEL-FREE TEST-TIME ADAPTATION

**Shahriar Rifat[†], Jonathan Ashdown[∗], Michael De Lucia[‡],
Ananthram Swami[‡] and Francesco Restuccia[†]**

[†] Northeastern University, United States
[‡] DEVCOM Army Research Laboratory, United States
[∗] Air Force Research Laboratory, United States

## ABSTRACT

Despite the success of Test-Time Adaptation (TTA), recent work has shown that adding relatively small adversarial perturbations to a limited number of samples in test data leads to significant performance degradation. Therefore, it is crucial to rigorously evaluate existing TTA algorithms against relevant threats and implement appropriate security countermeasures. Importantly, existing threat models assume test-time samples will be labeled, which is impractical in real-world scenarios. To address this gap, *we propose a new attack algorithm that does not rely on access to labeled test samples*, thus providing a concrete way to assess the security vulnerabilities of TTA algorithms. Our attack design is grounded in theoretical foundations and can generate strong attacks against different state of the art TTA methods. In addition, we show that existing defense mechanisms are almost ineffective, highlighting the need for further research on TTA security. Through extensive experiments on CIFAR10-C, CIFAR100-C, and ImageNet-C, we demonstrate that our proposed approach closely matches the performance of state-of-the-art attack benchmarks, even without access to labeled samples. In certain cases, our approach generates stronger attacks, e.g., more than 4% higher error rate on CIFAR10-C. Source code for the experiments is available at `https://github.com/Restuccia-Group/tta-adv.git`.

## 1 INTRODUCTION

Although Deep Neural Networks (DNNs) are expected to generalize beyond the training data distribution, it is nearly impossible to account for all the variations of the input distribution during training. As such, Test-Time Adaptation (TTA) is proposed to adapt a trained DNN using supervision from information extracted from unlabeled test data (Wang et al., 2020; Schneider et al., 2020; Goyal et al., 2022). Although designing loss functions from unlabeled data is challenging, TTA has been shown to outperform conventional DNN inference in various computer vision (CV) tasks, particularly in classification (Wang et al., 2020), object detection (Ruan & Tang, 2024), semantic segmentation (Wang et al., 2023), and multiple object tracking (Segu et al., 2023).

Despite the effectiveness of TTA, its robustness to adversarial samples is still unclear. In this threat model, imperceptible perturbations are tactically generated by adversaries to degrade the performance of TTA. Notice that this threat is fundamentally different from traditional evasion attacks (Goodfellow et al., 2014; Madry et al., 2018) or poisoning attacks Shafahi et al. (2018); Carlini & Terzis (2021), as adversaries do not need to modify specific samples or access the DNN training routine to induce erroneous predictions. Therefore, it is imperative to rigorously study adversarial threats to TTA. In particular, it is desirable to achieve appropriate utility-security trade-offs and implement defensive measures before deploying TTA algorithms in real-world high-stakes contexts.

The first work to investigate TTA security vulnerabilities was DIA (Wu et al., 2023), where it was shown that due to the transductive nature of learning in TTA, crafting an attack on some samples can potentially cause drastic performance degradation in non-malicious samples. The key issue is that DIA assumes access to the true labels of all test samples, thus crafting an adversarial perturbation that overestimates the difficulty of TTA attacks. Even if a malicious insider had access to the TTA system, as assumed in (Wu et al., 2023), only the prediction of the adapted DNN would be available to the adversary, and not the true labels. Under such an assumption, the attack proposed in (Wu

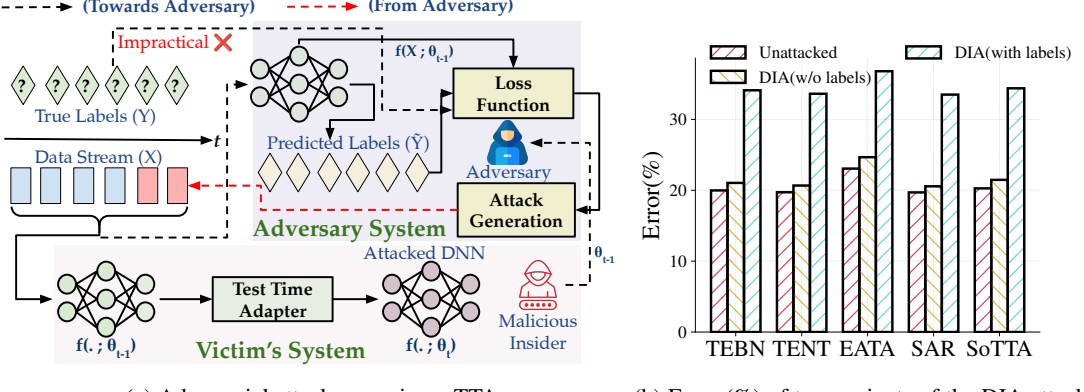

(a) Adversarial attack scenario on TTA

(b) Error (%) of two variants of the DIA attack on different TTA algorithms for CIFAR-10C.

Figure 1: 1a shows that even with a malicious insider involved, an adversary trying to attack a TTA system would not have access to true labels. Fig. 1b shows the efficacy of adversarial attacks on TTA for different cases of Distribution Invading Attack (DIA)(Wu et al., 2023) for CIFAR10-C. The attack efficacy also vanishes if access to true labels is not assumed.

et al., 2023) becomes weak, which gives a false sense of robustness of existing TTA algorithms. In stark contrast, in this work we show that even in the absence of true labels, an adversary can craft strong malicious samples that can degrade the performance of all the existing TTA algorithms. For the first time, our work provides a practical testbench to evaluate the adversarial robustness of TTA algorithms against malicious samples.

Our main contributions can be summarized as follows.

• We evaluate an impractical assumption in the current design of adversarial attacks on TTA – specifically, the access to test data labels – that renders it inapplicable in practical settings. Our study reveals that existing TTA methods demonstrate considerable robustness when this assumption is relaxed. This finding potentially opens new avenues for testing the security vulnerabilities of TTA in practical scenarios.

• We derive a formulation of the robust risk for TTA, which is more nuanced and helps us highlight the inherent security threats in existing TTA algorithms. From our formulation we design a novel attack algorithm termed as Feature Collapse Attack (FCA) that can generate strong attacks even in the absence of labels. Through extensive numerical experiments on three benchmark datasets and five TTA methods, we demonstrate that in practical label-free settings, existing TTA algorithms are almost equally or even more vulnerable in some cases against our proposed attack compared to an attack that assumes labelled data access.

• We show that existing defense mechanisms for robust TTA deployment are either ineffective or not directly applicable against our settings. This makes our proposed attack a strong threat requiring countermeasures to ensure robust and reliable deployment of TTA in real-world applications.

## 2 RELATED WORK

### 2.1 ROBUSTNESS OF TEST-TIME ADAPTATION

Significant effort has been made to improve the robustness of TTA in a variety of challenging scenarios, i.e., continual distribution shift (Wang et al., 2022), continuously changing smooth transition between domains (Press et al., 2024), when the distribution of labels changes in online test batches (Gong et al., 2022; Zhou et al., 2023), when these shifts happen concurrently (Yuan et al., 2023) and when some outlier samples are mixed in the data stream (Gong et al., 2024). However, the threat of the model adapting to potential adversarial samples mixed in the test data stream – as well as related defense mechanisms – are still underexplored. Two recent studies have shown that the predictive performance of certain samples under TTA can degrade without direct adversarial manipulation, simply by perturbing other parts of the online data batch in both white-box (Wu et al., 2023) and black-box settings (Cong et al., 2024). However, (Wu et al., 2023) assumes access to labeled samples to craft attacks, which is impractical in the TTA framework. On the other hand, the attack

proposed by (Cong et al., 2024) relies on mixing a portion of adversarial samples into consecutive test batches to achieve adversarial action. If the adversary can manipulate test batches intermittently or the model is reset frequently to prevent performance collapse, such threats become non-existent, potentially failing to reveal security concerns regarding TTA. As these threats gain attention, several defense mechanisms have been proposed to mitigate their impact, including entropy-based sample filtering combined with sharpness-aware minimization (Gong et al., 2024), median normalization (Park et al., 2024) and adversarial training (Wu et al., 2023). However, we show in Section 7.1 that these defense strategies are either ineffective or violate key assumptions of the original TTA framework. Ultimately, this highlights the need for more rigorous evaluation of TTA algorithms, and calls for the design and evaluation of robust TTA methods that can withstand adversarial action.

## 2.2 ADVERSARIAL MACHINE LEARNING

The literature on adversarial machine learning can be broadly divided into poisoning attacks on training data and evasion attacks on test data. Evasion attacks (Biggio et al., 2013; Carlini & Wagner, 2017) attempts to have the DNN misclassify inputs that have undergone minor perturbations. In a white-box setting, where an adversary can query the model and obtain gradients of a backward pass, this direction can be efficiently found through a gradient-based optimizer by maximizing the loss with respect to the input while constraining the perturbation. Data poisoning leverages the training process of DNNs to make it erroneously classify a benign sample by injecting some poisoned samples in the training set (Koh & Liang, 2017). There has been growing interest in adversarial attacks within low-label regimes and unsupervised settings. For example, (Kim et al., 2020) introduced instance-wise attacks without labels by maximizing contrastive loss for representation encoders, an approach further improved by (Fan et al., 2021) who incorporated high-frequency views. (Cemgil et al., 2020) developed adversarial attacks for variational autoencoders by maximizing the Wasserstein distance to the representations of clean samples. In the low-label regime, (Yang et al., 2023a) proposed generating adversarial attacks through adaptive weight regularization and knowledge distillation to improve the robustness of semi-supervised learning. However, none of these adversarial methods are directly applicable to our setting as they are not designed to deal with unlabeled test data in TTA scenarios.

## 3 PRELIMINARIES

### 3.1 TEST-TIME ADAPTATION

Without loss of generality, we cast TTA in the context of a multi-class classification problem. Let $\mathcal{X} \subset \mathbb{R}^d$ be the input space and the set of $C$ classes or output labels be denoted as $\mathcal{Y} = \{1 \ldots C\}$. Given a training data set from the source domain $\mathcal{D}^s := \left\{ (x_i^s, y_i^s)_{i=1}^{N_s} \right\}$ we learn $f_\theta : \mathcal{X} \to \mathbb{R}^C$ parameterized by the DNN parameter $\theta$ through iterative minimization of some loss function $\mathcal{L}(f(X^s; \theta), Y^s)$ (e.g., cross-entropy loss), where $X^s := \{x_i^s\}_{i=1}^{N_s} \subset \mathcal{X}$ and $Y^s := \{y_i^s\}_{i=1}^{N_s} \subset \mathcal{Y}$. The trained DNN is then deployed to perform inference on test dataset $\mathcal{D}^t := \left\{ (x_i^t, y_i^t)_{i=1}^{N_t} \right\}$ where the labels $Y^t := \{y_i^t\}_{i=1}^{N_t}$ are unknown. In conventional inference, it is assumed that $\mathcal{D}^s$ and $\mathcal{D}^t$ are sampled from the same distribution and the final predictions are calculated through $F_\theta = \arg\max_c \left[ f_\theta(x^t) \right]_c \subset \mathcal{Y}$. However, in real-world deployments, training and test data distributions may differ, which ultimately results in poor performance of $f_\theta$ on $X^t := \{x_i^t\}_{i=1}^{N_t}$ (Hendrycks & Dietterich, 2019). To address this issue, TTA dynamically adapts $f_\theta$ to the test data $X^t$ without supervision from $Y^t$, thus obtaining the adapted DNN $f_{\theta'}$. Unlike Unsupervised Domain Adaptation (UDA) where the entirety of the test samples is assumed to be available, in TTA $f_{\theta'}$ is obtained with the test data available in batch mode $X_B^t := \{x_i^t\}_{i=1}^{N_B}$ by solving

$$\theta'(X_B^t) = \arg\min_{\theta_A' \subset \theta'} \mathcal{L}_{TTA}\left( X_B^t; \theta' \right), \tag{1}$$

where $\theta'(X_B^t) = \theta_A' \cup \theta_B' \cup \theta_F$. Here, $\theta_A'$ indicates all adaptable parameters, $\theta_F$ are the fixed parameters and $\theta_B'$ are the normalization statistics $\{\mu, \sigma\}$ across different layers and $\mathcal{L}_{TTA}(\cdot)$ indicates the unsupervised TTA loss. Some existing works (Schneider et al., 2020; Gong et al., 2022) on TTA provide empirical evidence of performance improvement by only re-estimating the statistics of the Batch Normalization (BN) layers from test data. The absence of supervision is typically covered by

two unsupervised forms of losses, i.e., entropy minimization (Wang et al., 2021; Niu et al., 2022; Goyal et al., 2022) and invariance regularization (Wang et al., 2022; Nguyen et al., 2023).

## 3.2 NEURAL COLLAPSE (NC)

The NC phenomenon was first observed by (Papyan et al., 2020) in DNNs optimized with SGD, where some distinctive characteristics become increasingly apparent, particularly during the terminal phase of training. We formalize NC to ease the understanding of our attack algorithm. Let the $i$-th sample within class $c$ be denoted by $x_{i,c}$ and the last layer feature of a $k$-layer DNN as $\mathbf{h}_{i,c} = f_\theta^{k-1}(x_{i,c})$. The class mean and global sample mean are denoted by $\mu_c = \frac{1}{n}\sum_i \mathbf{h}_{i,c}$ and $\mu_G = \frac{1}{nC}\sum_{i,c} \mathbf{h}_{i,c}$, respectively, where $c = 1, \ldots, C$ and $n$ is the number of data points per class. We leverage the following three definitions directly from (Papyan et al., 2020).

**Variability Collapse:** For every class $c$, the within-class variation collapses to zero:

$$\mathbf{h}_{i,c} \to \mu_c \ \forall i \in [n], c \in [C] \tag{2}$$

**Equinorm:** Class feature mean vectors converge to equal distances from the global mean vector:

$$d(\mu_c, \mu_G) - d(\mu'_c, \mu_G) \to 0 \ \ \forall c, c' \in C \tag{3}$$

**Simplification to Nearest Neighbour Classifier (NNC):** The prediction of the DNN becomes equivalent to that of the NNC formed by non-centered class means:

$$\arg\max_{c'} \langle w_{c'}, h \rangle + b_{c'} \approx \arg\min_{c'} d(h, \mu_{\mathbf{c'}}) \tag{4}$$

## 3.3 UNDERSTANDING THE SECURITY VULNERABILITIES OF TTA

We start by describing the robust population risk $\mathcal{R}(\theta)$, which is a commonly used term in adversarial robustness (Zhang et al., 2019). Specifically, for a batch of test samples $X_B^t$ with true labels $Y_B^t$ and a predictive model $f_\theta(\cdot)$, the final predictions are obtained using $F_\theta(\cdot)$ as:

$$\mathcal{R}(\theta) = \mathbb{E}_{(X_B^t, Y_B^t)} \max_{\tilde{X}_B^t \in \mathcal{B}_p(X_B^t, \epsilon)} \mathbb{1}\left\{ F_\theta(\tilde{X}_B^t) \neq Y_B^t \right\} \tag{5}$$

where $\mathcal{B}_p(x, \epsilon) = \left\{ \tilde{x} \in \mathcal{X} : \|x - \tilde{x}\|_p \leq \epsilon \right\}$ and $\mathbb{1}(\cdot)$ is the indicator function. Following the formulation of (Yang et al., 2023b), the robust risk for a batch can be written as the sum of the natural risk $\mathcal{R}_{nat}(\theta)$ and the boundary risk $\mathcal{R}_{bdy}(\theta)$:

$$\mathcal{R}(\theta) = \mathcal{R}_{nat}(\theta) + \mathcal{R}_{bdy}(\theta) \tag{6}$$

in which the natural and boundary risk terms are defined as

$$\mathcal{R}_{nat}(\theta) = \mathbb{E}_{(X_B^t, Y_B^t)} \mathbb{1}\left\{ F_\theta(X_B^t) \neq Y_B^t \right\}$$
$$\mathcal{R}_{bdy}(\theta) = \mathbb{E}_{(X_B^t, Y_B^t)} \mathbb{1}\left\{ \exists \tilde{X}_B^t \in \mathcal{B}_p(X_B^t, \epsilon) \ : \ F_\theta(X_B^t) \neq F_\theta(\tilde{X}_B^t), \ F_\theta(X_B^t) = Y_B^t \right\}. \tag{7}$$

The natural risk represents the inherent failure of the DNN to fit certain data points, thus resulting in erroneous predictions. Conversely, boundary risk refers to the risk associated with regions in the manifold that are very close to the original data points but lead to incorrect predictions. In a traditional evasion attack, adding a bounded perturbation to samples potentially causes misclassification of those specific samples if it has high boundary risk. However, during the inference with TTA, the DNN is updated with the current data batch $\theta \to \theta'$, which can increase both the boundary risk $\mathcal{R}_{bdy}(\theta')$ and the natural risk $\mathcal{R}_{nat}(\theta')$ for the unperturbed samples if a small portion of the test samples are maliciously perturbed.

## 4 THREAT MODEL

We provide a description of the threat model considered in this paper. In our attack settings, the adversary has the ability to manipulate a small portion of the test samples within the entire batch

$X_B^t$, termed as *compromised samples* $X_{com}^t$. The attack is executed by adding imperceptible noise to those samples. Thus, the attacker targets the victim who gets a source model $f_\theta$ and tries to update it to $f_{\theta'}$ using TTA as in Equation 1. The attacker's objective is to indiscriminately degrade the predictions of the benign portion of the batch, $X_{B\setminus com}^t$, which can be defined as

$$\max_{\delta:\|\delta\|_p \leq \epsilon} \mathbb{L}\left[f\left(X_{B\setminus com}^t; \theta^*((X_{com}^t + \delta) \cup X_{B\setminus com}^t), Y_{B\setminus com}^t\right)\right], \tag{8}$$

where $\mathbb{L}[\cdot]$ denotes a loss function that directly relates to the prediction error. Similar to (Wu et al., 2023), we consider the worst-case scenario where the attacker has knowledge of the TTA algorithm and has white-box access to the latest updated model parameters provided by some malicious insider. However, unlike (Cong et al., 2024), we do not assume access to source data or true labels $Y_{B\setminus com}^t$ as in (Wu et al., 2023). *Notice that this is not a design choice but an inherent constraint of the TTA settings.* Additionally, we realistically do not assume that the attacker has any access to the training routine of the deployed DNN.

## 5   ATTACK DESIGN

The attacker's objective defined in Equation 8 essentially becomes the maximization of the robust population risk $\mathcal{R}(\theta')$ for the benign test samples. In (Wu et al., 2023), this is directly estimated by maximization of the cross entropy loss, which cannot be directly calculated without $Y_{B\setminus com}^t$. A viable alternative is to directly use the DNN prediction $\hat{Y}_{B\setminus com}^t$ or its soft prediction scores. However, as demonstrated earlier and in the experiments, this leads to an ineffective attack. Therefore, we provide an expression for the robust population risk that can be leveraged to successfully attack TTA and assess its robustness when true labels are not available. The following theorem provides an alternative expression for the risk in Equation 6. Details are provided in Appendix A1.

**Theorem 1** *For a batch of test samples, the robust population risk can be expressed as*

$$\mathcal{R}(\theta) = \mathbb{E}_{(X_B^t, Y_B^t)} \mathbb{1}\left\{F_\theta(X_B^t) \neq Y_B^t\right\} +$$
$$p\left(\exists \tilde{X}_B^t \in \mathcal{B}_p(X_B^t, \epsilon) : F_\theta(X_B^t) \neq F_\theta(\tilde{X}_B^t)\right) \cdot p\left(Y_B^t = F_\theta(X_B^t)|X_B^t\right). \tag{9}$$

This decomposition provides a more insightful look into the design of TTA attacks. Firstly, an adversary needs to increase the natural risk (first term) which needs access to true labels. In Section 5.1, we provide details on how this can be achieved from the assumptions of NC. Secondly, to maximize the boundary risk (second term), we need to add perturbations so as to increase the probability that the prediction changes after adding perturbation. Moreover, the second term needs to be multiplied with the probability assigned to the true class from the current DNN. These straightforward attack design directives are not readily available from Equation 6 and 7.

### 5.1   ESTIMATION OF ROBUST RISK

Using our decomposition of the robust risk, we design several loss functions that reliably estimate the loss term $\mathbb{L}(\cdot)$ in the attack objective without requiring labels. Below, we detail the loss functions motivated by Equation 9, which provides high-quality estimates of the objective function.

**Nearest Centroid Loss:** The natural risk (first term) is estimated using a loss function that we refer to as the nearest centroid loss $\mathcal{L}_{ncc}$:

$$\mathcal{L}_{ncc} = \sum_{x_i \in X_{B\setminus com}^t} d\left(f_\theta^{k-1}(x_i), \mu_{[\hat{C}=F_\theta(x_i)]}\right), \tag{10}$$

where $d(.)$ denotes some distance measure (e.g. cosine distance) and $\mu_{\hat{C}}$ denotes the class centroid vectors of the last layer features calculated from the activation values of the predicted classes $\hat{C}$ in a batch. This is based on the third assumption of NC described in Section 3.2 suggesting that an increased distance from the nearest centroid correlates with a higher likelihood of wrong prediction.

**Feature Collapse Loss:** As we are depending on the predicted labels for calculating the class centroids and the number of samples is small, these estimated class centroids are not guaranteed to be

reliable and might not provide proper guidance for the design of perturbation to increase the natural risk term. Hence, we incorporate the following term in our loss function:

$$\mathcal{L}_{col} = \sum_{x_i \in X^t_{B \setminus com}} d\left(f_\theta^{k-1}(x_i),\ \mu_{\hat{G}}\right). \tag{11}$$

The predicted global mean $\mu_{\hat{G}}$ is calculated by averaging out the activations in a batch. From the *Equinorm* assumption of NC, this moves the TTA update to a direction that compromises the distance of class means from the global sample mean. This acts as a regularizer that helps guide the attack design that is disrupted by the unreliable estimate of class centroids.

**Feature Scattering Loss:** The second term in our robust risk formulation is estimated leveraging the *Variability Collapse* assumption of NC. As the last layer activations converge to class mean vectors, the likelihood of overlapping regions with different class centroids diminishes. The following loss scatters the features away from forming tight clusters by maximizing the difference in the activation values before and after the TTA update:

$$\mathcal{L}_{sc} = \sum_{x_i \in X^t_{B \setminus com}} d\left(f_\theta^{k-1}(x_i),\ f_{\theta'}^{k-1}(x_i)\right) \tag{12}$$

Here, $f_{\theta'}$ is the DNN state after update on test data with malicious examples. According to our decomposition of robust risk in Equation 9, this loss term is multiplied by the probability of predicted class $p\left(\hat{C} == F_\theta(x_i)\right)$. Some deviation from the original value occurs when the predicted class is incorrect. An attacker would potentially benefit through a more accurate estimation of this term. Despite this non-ideal estimation, we can still craft highly-effective attacks against TTA as demonstrated in the experimental evaluation. The final loss function is:

$$\mathcal{L} = \mathcal{L}_{ncc} - \mathcal{L}_{col} + \mathcal{L}_{sc} \cdot p\left(\hat{C} == F_\theta(x_i)\right). \tag{13}$$

For the distance measure $d(.)$, we use cosine similarity as it avoids explicit normalization. $f_\theta^1(x_i)$

## 5.2 FEATURE COLLAPSE ATTACK (FCA): ALGORITHM DESIGN

The earlier loss functions are inspired from the assumptions of NC. As such, we name our attack algorithm as FCA. The process of adversarial perturbation generation in FCA is summarized in Algorithm 1. The algorithm is relatively easy to implement as it uses basic iterative projected gradient descent. At each iteration step, the perturbation vector is added to the compromised samples (line 6). Next, the DNN is updated using the TTA algorithm being considered (line 7). Then, using Equation 10-13, our proposed loss value is calculated. In line 10, gradients of the loss are calculated. Finally, the perturbation is updated using adversarial learning rate $\alpha$ and clipped to $(-\epsilon, \epsilon)$ to maintain the $l_\infty$ norm constraint. The perturbation $\delta$ is also constrained such that the pixel values always remain within the $[0, 1]$ range.

## 6 EXPERIMENTAL SETUP

**Dataset and Architecture** We leverage three primary benchmark datasets typically used for TTA performance evaluation, i.e., CIFAR10-C, CIFAR100-C, and ImageNet-C. We directly obtain the CIFAR10-C and CIFAR100-C test dataset from Robustbench (Croce et al., 2020). For ImageNet-C, we use the provided data by (Hendrycks & Dietterich, 2019). These datasets are modified versions of the original test data by 15 different synthetically generated corruption of 5 severity levels. For our experiments, we use ResNet-32[1] for CIFAR10-C and CIFAR100-C datasets and ResNet-50 for ImageNet-C[2]. For the two corrupted CIFAR datasets, all 15 corruptions for 5 severity levels are used. For Imagenet-C, results are reported only for corruption of severity level 3.

**Baseline TTA Methods.** We evaluate 5 TTA methods as victim algorithms whose robustness is tested against FCA. In line with the previous studies (Wu et al., 2023; Park et al., 2024), we select TTA methods that focus on updating batch statistics or the affine parameters of BN layers. Test-time Normalization (TeBN) (Nado et al., 2020) updates BN statistics for each test batch, while

---

[1]Model defination and trained weights are directly obtained from `https://github.com/chenyaofo/pytorch-cifar-models`.

[2]We have used the model definition and weights from torchvision(resnet50-v2)

---

**Algorithm 1:** FCA Algorithm

---

1: **Input:** Test batch $X_B^t = X_{com}^t \cup X_{B \setminus com}^t$ ; Adversarial learning rate $\alpha$; Perturbation constraint $\epsilon$; Model paramaters before adaptation $\theta$; Iteration steps for attack $n$;

2: **Initialize:** Adversarial perturbation $\delta = \mathbf{0.5}^{\left| X_{com}^t \right| \times c \times h \times w} \rightarrow (c, h, w)$ represent the channel, height, and width of an input sample

3: **Output:** Adversarial perturbation vector $\delta$

4: Make a separate copy of $\theta$

5: **for** step = 1, 2, ....., n **do**

6:     $X_B^t = \{X_{com}^t + \delta\} \cup X_{B \setminus com}$

7:     Update $\theta \rightarrow \theta'$ [Eq. 1]

8:     Calculate $f_\theta^{k-1}(X_{B \setminus com}^t)$ and $f_{\theta'}^{k-1}(X_{B \setminus com}^t)$

9:     Calculate $\mathcal{L}(X_{B \setminus com}; \cdot)$ [Eq. 13]

10:     Calculate $grad = \nabla_{\delta_{com}} \left( \frac{\mathcal{L}(X_{B \setminus com}^t; \cdot)}{\left| X_{B \setminus com}^t \right|} \right)$

11:     Calculate $\delta \leftarrow clip\left((\delta + \alpha \cdot sign(grad)), (-\epsilon, \epsilon)\right)$

12:     Calculate $\delta \leftarrow clip\left((X_{com}^t + \delta_{com}), (0, 1)\right) - X_{com}^t$

13: **end for**

14: **Return:** $X_{com}^t + \delta$

---

Test-time Entropy Minimization (TENT) (Wang et al., 2021) adjusts the affine parameters in BN layers through entropy minimization. Efficient Anti-forgetting Test-time Adaptation (EATA) (Niu et al., 2022) enhances sample-efficient entropy minimization and incorporates a Fisher regularizer to prevent knowledge loss from the pre-trained model. Sharpness-aware and Reliable Optimization (SAR) (Niu et al., 2023) utilizes BN layers and sharpness-aware minimization to mitigate the adverse effects of large gradients, and Screening-out Test-time Adaptation (SoTTA) (Gong et al., 2024) employs sample filtering and sharpness-aware minimization.

**Evaluation Setup** To benchmark the performance of FCA, we consider each test batch as a *trial*. For a given test batch, we randomly select some data samples as compromised ones and add perturbations generated by one of the baseline attack methods as well as FCA. We assume the scenario where the DNN is updated by the considered TTA algorithm up to the $t-1$ trial and the adversary attacks the update of trial $t$, $f_{\theta'_t}(\cdot)$ with access to model parameters $f_{\theta'_{t-1}}(\cdot)$. We evaluate the effectiveness of FCA by its average error rate increase on the benign (unperturbed) samples compared to normal adaptation without attack across all trials. Unless otherwise specified, we use a test batch size of 200 for each trial where 20% samples are selected as compromised ones, adversarial learning rate $\alpha = 2/255$, perturbation constraint $\epsilon = 8/255$ and iteration steps for attack to be 100.

## 7 EXPERIMENTAL RESULTS

We compare our proposed FCA against five state-of-the-art TTA methods across three benchmark datasets. For attack baselines, we include three benchmarks: the Distributionally Informed Attack (DIA) (Wu et al., 2023), its pseudo-label variant DIA(PL), and TePA (Cong et al., 2024), originally proposed as a test-time poisoning attack. In our experiments, we adapt TePA's loss formulation under our threat model and within the context of each TTA method. Pseudo labels can be obtained using the current model's prediction probabilities (soft PL) or the argmax of these predictions (hard PL) for benchmarking. Since both variants result in similarly low attack efficacy, we use hard pseudo labels throughout and refer to them as PL for brevity. Notice that the original DIA setup assumes access to the true labels of the data, which are unavailable at test time. From Table 1, we observe that in CIFAR-10C, the proposed FCA consistently increases the error rate by more than 4% for TENT and TeBN and by similar margin in other TTA algorithms. However, without access to the true labels, DIA is almost ineffective. This is because the predicted labels are inherently noisy due to the distribution change in the data. For CIFAR-100C and ImageNet-C, our proposed attack mechanism also performs very close to the DIA benchmark. Across these three datasets, we observe that with the increasing number of unique classes, our attack is still very effective but weakens slightly compared

Table 1: Performance (% error) comparison of different TTA methods.

| Dataset | Attack Method | TTA Method | | | | |
|---|---|---|---|---|---|---|
| | | TeBN | TENT | EATA | SAR | SoTTA |
| **CIFAR10-C** | w/o Attack | 19.98 | 19.72 | 23.05 | 19.70 | 20.27 |
| | DIA | 34.11 | 33.62 | 36.80 | 33.50 | 34.39 |
| | DIA (PL) | 21.04 | 20.67 | 24.23 | 21.03 | 20.98 |
| | TePA | 21.11 | 20.81 | 24.27 | 21.01 | 21.04 |
| | FCA | **38.87** | **37.95** | **40.15** | **37.33** | **38.37** |
| **CIFAR100-C** | w/o Attack | 51.44 | 49.39 | 52.56 | 49.39 | 50.01 |
| | DIA | **64.21** | **62.64** | 63.76 | **62.57** | **63.45** |
| | DIA (PL) | 52.21 | 52.67 | 53.69 | 51.03 | 52.07 |
| | TePA | 52.29 | 52.68 | 53.81 | 51.09 | 52.12 |
| | FCA | 64.03 | 62.21 | **63.97** | 62.12 | 63.23 |
| **ImageNet-C** | w/o Attack | 43.78 | 42.33 | 43.38 | 43.37 | 42.82 |
| | DIA | **51.54** | **50.65** | **49.54** | **52.06** | **51.23** |
| | DIA (PL) | 46.29 | 45.11 | 44.88 | 44.77 | 44.79 |
| | FCA | 51.06 | 50.63 | 49.06 | 50.22 | 50.33 |

to DIA. We believe this is due to the effect of the NC assumptions getting slightly relaxed as the number of classes increases.

**Performance Evaluation in grey box settings** For the results reported in the main section, we evaluated the effectiveness of FCA under a worst-case scenario where the adversary has access to the latest DNN weights during model updates. Although this setting reveals the worst-case scenario, it may not always be practical for real-world deployment. To assess the performance of FCA under more restrictive grey-box settings, we relaxed the initial assumptions. We assume a threat model where the adversary only has access to the source DNN weights and architecture used for TTA, but not the latest updated parameters. The evaluation results, presented in Table 2, show that our proposed attack mechanism remains highly effective in these settings, with only a $\sim 2\%$ reduction in attack effectiveness across different TTA methods.

Table 2: Performance (% error) comparison in grey box settings

| Dataset | Attack Method | TTA Method | | | | |
|---|---|---|---|---|---|---|
| | | TeBN | TENT | EATA | SAR | SoTTA |
| **CIFAR10-C** | w/o Attack | 19.98 | 19.72 | 23.05 | 19.70 | 20.27 |
| | DIA | 33.22 | 31.77 | 33.53 | 31.03 | 32.09 |
| | DIA (PL) | 20.91 | 20.63 | 24.23 | 21.05 | 20.87 |
| | TePA | 21.11 | 20.81 | 24.27 | 21.01 | 21.04 |
| | FCA | **36.03** | **35.98** | **37.41** | **34.83** | **36.29** |
| **CIFAR100-C** | w/o Attack | 51.44 | 49.39 | 52.56 | 49.39 | 50.01 |
| | DIA | **61.03** | **60.21** | 60.51 | **60.11** | **62.34** |
| | DIA (PL) | 52.13 | 52.58 | 53.66 | 50.81 | 52.05 |
| | TePA | 52.29 | 52.68 | 53.81 | 51.09 | 52.12 |
| | FCA | 60.22 | 60.04 | **60.58** | 69.91 | 62.07 |
| **ImageNet-C** | w/o Attack | 43.78 | 42.33 | 43.38 | 43.37 | 42.82 |
| | DIA | **49.36** | **48.88** | **47.15** | **50.11** | **49.38** |
| | DIA (PL) | 45.23 | 44.87 | 43.15 | 43.39 | 42.92 |
| | FCA | 49.03 | 48.55 | 47.13 | 50.07 | 49.21 |

**Effect of Sample Size** In Table 4, the performance of the FCA and DIA attacks is reported for different numbers of maliciously perturbed samples on the CIFAR-10C dataset. Interestingly, FCA exhibits a higher error rate across all TTA benchmarks compared to DIA, indicating that it does not have any extra sensitivity to number of malicious samples compared to the DIA. To evaluate the

efficacy of the attack with different batch sizes, Table 3 reports the performance of FCA on three different batch sizes for the CIFAR-100C dataset. The CIFAR-100C dataset was chosen due to its larger number of classes compared to CIFAR-10C, which could reveal any potential sensitivity of our designed attack as class centroid estimation becomes unreliable with smaller batch sizes and higher number of unique classes. However, we observe that the performance of FCA does not decrease significantly compared to DIA for low sample sizes.

**Effect of Different Loss Components of FCA** To analyze the efficacy of different FCA loss terms, we evaluate the performance of FCA on CIFAR-10C in the common attack parameter settings. From Table 5, it can be observed that feature scattering loss is the most influential in the efficacy of attack. However, each of the loss terms has a notable effect on the effectiveness of our porposed attack.

Table 3: Performance (% error) across different batch sizes on CIFAR-100C

| Batch | TTA Method (DIA/FC) | | | | |
|---|---|---|---|---|---|
| Size | TeBN | TENT | EATA | SAR | SoTTA |
| 64 | 70.93 / 68.98 | 70.86 / 69.03 | 71.04 / 69.22 | 69.03 / 68.55 | 69.21 / 67.84 |
| 128 | 64.93 / 64.24 | 63.96 / 62.30 | 65.01 / 64.88 | 63.12 / 62.74 | 63.44 / 62.29 |
| 200 | 64.21 / 64.03 | 62.64 / 62.21 | 63.76 / 63.97 | 62.57 / 62.21 | 63.45 / 63.23 |

Table 4: Performance (% error) with different numbers of malicious samples on CIFAR-10C

| # Malicious | TTA Method (DIA/FC) | | | | |
|---|---|---|---|---|---|
| Samples | TeBN | TENT | EATA | SAR | SoTTA |
| 10 (5%) | 23.33 / 25.11 | 23.02 / 24.81 | 24.11 / 26.77 | 23.03 / 24.21 | 23.02 / 24.21 |
| 20 (10%) | 26.33 / 31.22 | 25.77 / 29.98 | 27.64 / 32.24 | 25.44 / 29.67 | 36.75 / 30.11 |
| 40 (20%) | 34.11 / 38.77 | 33.62 / 37.95 | 36.80 / 40.15 | 33.50 / 37.33 | 34.39 / 38.37 |
| 80 (40%) | 45.33 / 49.81 | 43.77 / 48.21 | 47.03 / 52.31 | 43.98 / 47.75 | 43.21 / 47.33 |

## 7.1 LIMITATION OF EXISTING DEFENSE MECHANISM

**Sharpness Aware Learning** Sharpness-aware learning (Niu et al., 2023) aims to enhance the stability of model parameters by steering them towards a flat minimum on the loss surface. This method is grounded in the idea that a flat minimum is preferable for model robustness, particularly when dealing with noisy or large gradients. However, as shown in Fig. 2a,

Table 5: Performance (% error) on different FCA variants on CIFAR-10C

| FC Variant | TeBN | TENT | EATA | SAR | SoTTA |
|---|---|---|---|---|---|
| w/o attack | 19.98 | 19.72 | 23.05 | 19.7 | 20.27 |
| $\mathcal{L}_{nc}$ | 21.21 | 20.93 | 22.01 | 21.03 | 20.87 |
| $\mathcal{L}_{col}$ | 23.22 | 23.01 | 24.55 | 23.97 | 22.66 |
| $\mathcal{L}_{sc}$ | 34.33 | 33.11 | 36.24 | 35.18 | 34.13 |
| $\mathcal{L}_{col} + \mathcal{L}_{nc}$ | 23.24 | 23.03 | 24.61 | 23.95 | 22.67 |
| $\mathcal{L}_{sc} + \mathcal{L}_{col}$ | 35.91 | 35.11 | 37.89 | 36.22 | 34.74 |
| $\mathcal{L}_{sc} + \mathcal{L}_{nc}$ | 37.91 | 37.03 | 38.94 | 36.54 | 37.26 |
| Final $\mathcal{L}$ | 38.87 | 37.95 | 40.15 | 37.33 | 38.37 |

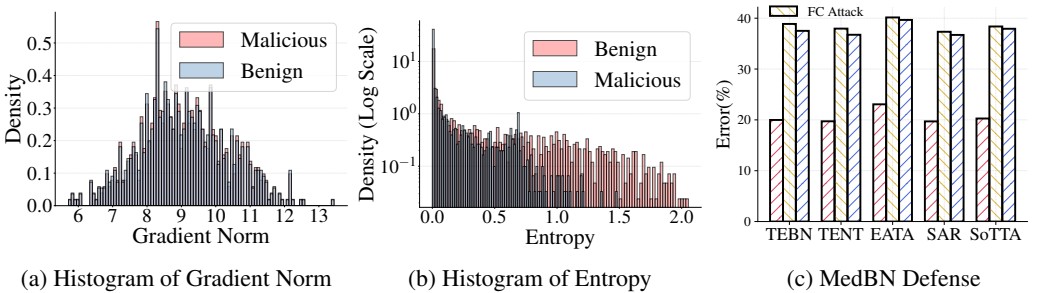

(a) Histogram of Gradient Norm  (b) Histogram of Entropy  (c) MedBN Defense

Figure 2: Limitation of different defense algorithms. The plots are generated with CIFAR10-C with the default attack parameters setting.

adversarial samples generated by our attack mechanism exhibit gradients similar to those of normal samples. Additionally, there is a high concentration of malicious samples in the region where gradient norms are small. This suggests that sharpness-aware learning may not be an effective strategy for mitigating the impact of adversarial data. From, Table 1, it is also evident that methods that incorporate sharpness aware learning (SoTTA) are not more robust than other TTA scheme that does not involve such a mechanism.

**Sample Filtering Scheme** As a defense mechanism, (Gong et al., 2024) proposed filtering out high entropy samples. However, the histogram in Figure 2b reveals that the entropy values of adversarial samples and normal samples are similarly distributed. In fact, most adversarial samples crafted by our method are concentrated in the low entropy region. Consequently, TTA methods that use a simple entropy-based filtering scheme, as suggested by (Niu et al., 2022; Gong et al., 2024), do not demonstrate significant robustness in our experiments.

**Robust Statistics Estimation** (Wu et al., 2023) proposed using robust BN statistics by treating source statistics as a prior and updating them with test data statistics using a momentum term. Since source data statistics remain unaffected by adversarial attacks, it improves robustness. However, overemphasizing source statistics can hinder the extraction of information from test samples, affecting TTA performance. Therefore, selecting the appropriate momentum value, which balances this trade-off, is crucial. However, the optimal momentum value varies across different TTA methods, as shown in Table 6, making the defense mechanism highly dependent on this hyper parameter choice.

A recent defense mechanism proposed by (Park et al., 2024) demonstrates that Median Batch Normalization (MedBN), being robust to outliers, is a viable alternative to BN. However, incorporating median normalization or its variants only complicates the generation of adversarial examples. A critical flaw in this defense mechanism is that it only superficially increases the difficulty of generating adversarial samples without providing substantial robustness to the TTA method. An adversary can easily bypass this defense by crafting adversarial perturbations using a traditional BN layer instead of the MedBN layer. Figure 2c illustrates that even when the victim TTA adapters are equipped with MedBN, it is still largely ineffective in preventing performance degradation in the event of attack.

Table 6: Performance (% error) for different momentum values for robust statistics estimation on CIFAR10-C

| Momentum | TeBN | TENT | EATA | SAR | SoTTA |
|---|---|---|---|---|---|
| 0 | 38.87 | 37.95 | 40.15 | 37.33 | 38.37 |
| 0.4 | 37.70 | 35.26 | **35.33** | 35.22 | 34.67 |
| 0.6 | 36.21 | 34.48 | 35.36 | **35.14** | **33.67** |
| 0.8 | **34.22** | **33.56** | 35.33 | 35.03 | 33.71 |

## 8 CONCLUSIONS

Our study highlights significant vulnerabilities in current TTA methods when faced with adversarial attacks, especially under practical conditions where test sample labels are not accessible. We challenge the prevailing assumptions in existing threat models and demonstrate that many TTA methods are more robust than previously thought when these assumptions are relaxed. However, our newly proposed attack algorithm, which does not rely on labeled test samples, reveals that TTA methods still possess inherent security risks. Our extensive experiments on benchmark datasets confirm that our approach can generate strong attacks, sometimes surpassing state-of-the-art benchmarks which assume access to labels. We find that existing defense mechanisms are largely ineffective against these types of attacks, underscoring the need for further research and development in this area.

## ACKNOWLEDGMENT OF SUPPORT

This work has been funded in part by the National Science Foundation under grants ECCS-2229472 and CNS-2312875, by the Air Force Office of Scientific Research under contract number FA9550-23-1-0261, by the Office of Naval Research under award number N00014-23-1-2221.

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

## A1    DETAILS OF THEOREM 1

Here, we provide a detailed proof of the decomposition of the robust risk given in Eq. 9.

**Theorem 1.** For a DNN $f_\theta$, parameterized by $\theta$, the robust risk $\mathcal{R}(\theta)$ for a batch of samples $(X_B^t, Y_B^t)$ can be written as:

$$\mathcal{R}(\theta) = \mathbb{E}_{(X_B^t, Y_B^t)} \mathbb{1}\left\{ F_\theta(X_B^t) \neq Y_B^t \right\} +$$
$$p\left( \exists \tilde{X}_B^t \in \mathcal{B}_p(X_B^t, \epsilon) : F_\theta(X_B^t) \neq F_\theta(\tilde{X}_B^t) \right) \cdot p\left( Y_B^t = F_\theta(X_B^t) \mid X_B^t \right) \tag{A1}$$

**Proof.** From Eq. 6, $\mathcal{R}(\theta) = \mathcal{R}_{nat}(\theta) + \mathcal{R}_{bdy}(\theta)$. where, $\mathcal{R}_{nat}(\theta) = \mathbb{E}_{(X_B^t, Y_B^t)} \mathbb{1}\{F_\theta(X_B^t) \neq Y_B^t\}$ and $\mathcal{R}_{bdy}(\theta) = \mathbb{E}_{(X_B^t, Y_B^t)} \mathbb{1}\left\{ \exists \tilde{X}_B^t \in \mathcal{B}_p(X_B^t, \epsilon) : F_\theta(X_B^t) \neq F_\theta(\tilde{X}_B^t), F_\theta(X_B^t) = Y_B^t \right\}$.

Since,

$$\mathcal{R}_{bdy}(\theta) = \mathbb{E}_{(X_B^t, Y_B^t)} \mathbb{1}\left\{ \exists \tilde{X}_B^t \in \mathcal{B}_p(X_B^t, \epsilon) : F_\theta(X_B^t) \neq F_\theta(\tilde{X}_B^t), F_\theta(X_B^t) = Y_B^t \right\}$$

$$= \mathbb{E}_{(X_B^t, Y_B^t)} \mathbb{1}\left\{ \exists \tilde{X}_B^t \in \mathcal{B}_p(X_B^t, \epsilon) : F_\theta(X_B^t) \neq F_\theta(\tilde{X}_B^t) \right\} \cdot \mathbb{1}\left\{ F_\theta(X_B^t) = Y_B^t \right\}$$

$$= \mathbb{E}_{X_B^t} \left[ \mathbb{1}\left\{ \exists \tilde{X}_B^t \in \mathcal{B}_p(X_B^t, \epsilon) : F_\theta(X_B^t) \neq F_\theta(\tilde{X}_B^t) \right\} \cdot \mathbb{E}_{(Y_B^t | X_B^t)} \mathbb{1}\left\{ F_\theta(X_B^t) = Y_B^t \right\} \right]$$

$$[\because \text{Law of Iterated Expectation}]$$

$$= \mathbb{E}_{X_B^t} \left[ \mathbb{1}\left\{ \exists \tilde{X}_B^t \in \mathcal{B}_p(X_B^t, \epsilon) : F_\theta(X_B^t) \neq F_\theta(\tilde{X}_B^t) \right\} \cdot p\left( Y_B^t = F_\theta(X_B^t) \mid X_B^t \right) \right]$$

$$= \mathbb{E}_{X_B^t} \left[ \mathbb{1}\left\{ \exists \tilde{X}_B^t \in \mathcal{B}_p(X_B^t, \epsilon) : F_\theta(X_B^t) \neq F_\theta(\tilde{X}_B^t) \right\} \right] \cdot \mathbb{E}_{X_B^t} \left[ p\left( Y_B^t = F_\theta(X_B^t) \mid X_B^t \right) \right]$$

$$= p\left( \exists \tilde{X}_B^t \in \mathcal{B}_p(X_B^t, \epsilon) : F_\theta(X_B^t) \neq F_\theta(\tilde{X}_B^t) \right) \cdot p\left( Y_B^t = F_\theta(X_B^t) \mid X_B^t \right)$$

Thus the equality holds.

## A2    EFFECT OF DNN ARCHITECTURE ON FCA

To examine whether the source DNN architecture significantly impacts TTA vulnerabilities, we evaluated the performance of FCA on the MobileNet family of DNNs, specifically using MobileNet-v2 as the source DNN. We assessed the performance of five baseline TTA methods across three benchmark datasets and report the results in Table 7. Our findings show that TTA methods exhibit similar vulnerabilities on CIFAR-10C and CIFAR-100C datasets as those observed with ResNet variants. However, for the ImageNet-C benchmark, MobileNet-v2 proved to be even more vulnerable, with performance degradation under FCA being $\sim 10\%$ greater compared to the ResNet-50 results.

## A3    PERFORMANCE EVALUATION FOR ADVERSARIALLY TRAINED MODELS

A potential defense against the vulnerabilities highlighted by FCA is to proactively use an adversarially trained source DNN. To evaluate this, we utilized the adversarially trained WideResNet-28 with an $l_\infty$ budget ($\epsilon_\infty = 8/255$) from Robustbench Croce et al. (2020) by Wu et al. (2020), and assessed its performance on CIFAR10-C and CIFAR-100C benchmark datasets. The results are reported in Table 8. Adversarially trained DNNs are highly effective against FCA when the same perturbation is used for both crafting adversarial examples and training the source DNN. However, with a different perturbation budget, such as an $l_2$ norm constraint of ($\epsilon_\infty = 8/255$), FCA can still degrade performance by approximately 4%. Furthermore, for the CIFAR-100C dataset, adversarially trained source DNNs result in more than a 10% increase in error rate during adaptation with benign data. This is unexpected, as TTA is generally intended to handle online data batches without adversarial perturbation, raising concerns about the robustness-utility trade-off in deploying adversarially

Table 7: (% Error) comparison on MobileNet architectures.

| Dataset | Attack Method | TTA Method | | | | |
|---|---|---|---|---|---|---|
| | | TeBN | TENT | EATA | SAR | SoTTA |
| **CIFAR10-C** | w/o Attack | 21.04 | 21.55 | 23.94 | 20.93 | 19.91 |
| | DIA | 35.54 | 35.11 | 35.56 | 34.44 | 34.48 |
| | DIA (PL) | 22.07 | 22.64 | 24.87 | 21.55 | 21.03 |
| | TePA | 22.44 | 22.61 | 24.81 | 21.63 | 21.15 |
| | FCA | **39.33** | **38.34** | **40.01** | **38.07** | **38.09** |
| **CIFAR100-C** | w/o Attack | 45.55 | 44.81 | 45.83 | 44.63 | 43.84 |
| | DIA | **57.13** | **55.45** | 57.03 | **56.14** | **55.93** |
| | DIA (PL) | 46.67 | 46.55 | 47.03 | 46.41 | 45.59 |
| | TePA | 46.74 | 46.51 | 46.98 | 46.55 | 45.71 |
| | FCA | 56.88 | 55.19 | **57.21** | 56.04 | 55.45 |
| **ImageNet-C** | w/o Attack | 54.2 | 52.97 | 53.78 | 52.83 | 51.29 |
| | DIA | **71.56** | **70.37** | **70.45** | **70.87** | 68.55 |
| | DIA (PL) | 57.5 | 56.44 | 56.29 | 55.31 | 54.92 |
| | FCA | 71.44 | 70.01 | 70.31 | 70.15 | **70.22** |

Table 8: (% Error) comparison on adversarially trained models.

| Dataset | Evaluation Setup | TTA Method | | | | |
|---|---|---|---|---|---|---|
| | | TeBN | TENT | EATA | SAR | SoTTA |
| **CIFAR10-C** | Unattacked(Standard) | 17.14 | 16.98 | 19.21 | 16.88 | 16.42 |
| | Unattacked(Adv trained) | 19.21 | 16.22 | 18.44 | 17.91 | 15.40 |
| | FCA ($\epsilon_\infty = 8/255$) | 21.44 | 18.01 | 20.25 | 19.83 | 17.17 |
| | FCA ($\epsilon_2 = 0.5$) | 23.45 | 20.14 | 22.03 | 21.55 | 19.03 |
| **CIFAR100-C** | Unattacked(Standard) | 31.27 | 30.91 | 31.87 | 30.9 | 29.3 |
| | Unattacked(Adv trained) | 42.04 | 41.59 | 42.14 | 41.51 | 41.04 |
| | FCA ($\epsilon_\infty = 8/255$) | 43.01 | 42.57 | 20.25 | 42.79 | 41.85 |
| | FCA ($\epsilon_2 = 0.5$) | 46.44 | 45.22 | 44.55 | 45.01 | 44.76 |

trained DNNs. Additionally, adversarial training is known to reduce accuracy on clean data Zhang et al. (2019); Tsipras et al. (2018). Thus, further scrutiny is required to develop computationally lightweight test-time defenses that are effective against FCA without impairing TTA performance on clean or benign samples from different domains.

Table 9: (% Error) comparison on robust models

| Dataset | Evaluation Setup | TTA Method | | | | |
|---|---|---|---|---|---|---|
| | | TeBN | TENT | EATA | SAR | SoTTA |
| **CIFAR10-C** | Unattacked(AugMix) | 15.37 | 14.81 | 16.47 | 14.53 | 14.11 |
| | FCA ($\epsilon_\infty = 8/255$ | 24.33 | 23.21 | 24.98 | 23.05 | 22.87 |
| **CIFAR100-C** | Unattacked(AugMix) | 29.34 | 28.77 | 30.21 | 28.55 | 27.87 |
| | FCA | 36.41 | 35.22 | 37.02 | 34.75 | 34.28 |

## A4 PERFORMANCE EVALUATION FOR ROBUST MODELS

To further understand how the robustness of the source DNN influences FCA, we analyzed the performance of FCA against source DNNs known for their robustness to distribution shifts. Specifically, we utilized the WideResNet-28 model trained with AugMix Hendrycks et al. (2019) from Robustbench Croce et al. (2020), and the evaluation results are presented in Table 9. While AugMix-trained

models are effective in enhancing robustness against various distribution shifts, they remain highly vulnerable to FCA when deployed for TTA.

