# OpenReview forum: "On the Adversarial Vulnerability of Label-Free Test-Time Adaptation"
_ICLR.cc/2025/Conference — ICLR 2025 Poster_

### Official Review · Reviewer_AcHd · 2024-10-31

**Soundness:** 3
**Presentation:** 3
**Contribution:** 3
**Rating:** 3
**Confidence:** 5

**Summary:**

To address the distribution shift of deep neural networks (DNNS), test-time adaptation (TTA) updates the model parameters using test samples. However, recent work reveals that TTA algorithms are vulnerable to adversarial attacks. This paper proposes a novel attack named Feature Collapse Attack (FCA). To be specific, the adversarial generated through FCA can be mixed with benign samples, thereby degrading the performance of the target model. A main feature of FCA is that it does not need the label of the test samples.

**Strengths:**

- This paper is well-written and easy-to-follow
- TTA algorithms are the key developing trend of AI, and the evaluation of the adversarial robustness of TTA algorithms is timely and necessary.

**Weaknesses:**

- The threat model of this paper is impractical.
- The experiments are insufficient.
- Important baseline attack is missing.

**Questions:**

Thanks for submitting the paper to ICLR 2025!

This paper aims to address a timely and important problem, i.e., evaluating the adversarial robustness of TTA algorithms. As the authors claimed, launching adversarial attacks against TTA algorithms is not a novel question, but at an early stage. Meanwhile, mounting adversarial attacks against TTA algorithms is not trivial, as the capability of the attacker is limited. The authors demonstrate the effectiveness of their proposed FCA attack. However, I have the following questions.

- My biggest concern is the threat model of this paper.

     - One of the main claims of this article is the practicability of the attacks. The paper pointed out that previous attacks like DIA need the label of the test sample. The reviewer agrees that it is very valuable to design a label-free attack. However, in Section 4, the authors claimed that the attacker has white-box access to the latest updated model parameters provided by some malicious insider. In my opinion, this assumption is impractical. First, who is the malicious insider? Can the authors provide a real-world example?
     - In the threat model of (Cong et al., 2024), they proposed a black-box attack named TePA. TePA does not need access to the parameters of the updated model, which is more practical.
     - The authors also claimed that "existing threat models assume test-time samples will be labeled", and TePA (Cong et al., 2024) relies on access to the entire source dataset to train a substitute model. Actually, TePA also is a label-free attack, and it does not need access to the entire source dataset. It only needs a surrogate dataset that has a similar distribution of the source dataset. I suggest that the authors reorganize the introduction of related work.

- In experiments like Table 1, this paper does not compare the attack success rate with TePA (Cong et al., 2024).
- This paper considers five target TTA methods, which all need the test samples to be fed in a batch-by-batch manner. However, some TTA methods like TTT [R1] can accept only one sample to adjust the model. I am curious if FCA can attack such kinds of TTA methods.
- The proposed attack seems not to be stealthy. In other words, FCA needs a high poisoning rate (Table 3).
- How many batches does FCA need to effectively decrease the target model's performance?
- In (Cong et al., 2024). they also proposed several defenses, which are not discussed in this paper.

Reference

[R1] Test-Time Training with Self-Supervision for Generalization under Distribution Shifts. ICML 2020.

---

> ### Author Response · Authors · 2024-11-24
>
> > One of the main claims of this article is the practicability of the attacks. The paper pointed out that previous attacks like DIA need the label of the test sample. The reviewer agrees that it is very valuable to design a label-free attack. However, in Section 4, the authors claimed that the attacker has white-box access to the latest updated model parameters provided by some malicious insider. In my opinion, this assumption is impractical. First, who is the malicious insider? Can the authors provide a real-world example?
>
> We thank you for giving us the opportunity to clarify this crucial point. Please see the common response.
>
> > In the threat model of (Cong et al., 2024), they proposed a black-box attack named TePA. TePA does not need access to the parameters of the updated model, which is more practical.
>
> Thank you for pointing out the TePA attack. We remark that in TePA, the adversary needs to feed poisoned samples across consecutive test batches to have effect. One key distinction between our evaluation of attack's effectiveness compared to TePA is that we only consider how non-perturbed benign samples are hampered in a certain batch for evaluation. In stark opposition with TePA, we consider every new batch as an independent trial, which is a different and more challenging scenario. For example, to degrade performance using the TENT [1] on fog corruption for the CIFAR-100 dataset by only $\sim$ 2.6\%, TePA needs adaptation on 40 batches (batch size of 200), with 20\% of the samples poisoned. If the victim performs TTA in a episodic manner (i.e., the DNN is reset to original state), TePA will become less effective. Moreover, considering threats in our scenario is more in line with recent work  [2] which strongly advocates resetting the model to prevent performance degradation.
>
>
> Additionally, to evaluate  TePA in our setting, we craft adversarial samples using the trained substitute black-box model provided by the authors by using the same loss function and optimization process (Diverse Input FGSM instead of PGD). We added the  results in Table 1 of the updated manuscript.  Table 1 shows that  TePA performs nearly as well as the weakest benchmark in the paper, which is \gls{dia} with pseudo-labels (PL). Indeed, we point out that taking the model’s soft predictions as pseudo-labels is equivalent to entropy maximization with an inverted sign. Hence, maximizing entropy also maximizes deviation from the model's own predictions (pseudo-labels), resulting in similar performance for both benchmarks. The minor deviation observed is due to differences in the optimization process (e.g., iterative PGD vs. diverse input FGSM) and the surrogate models.
>
> > The authors also claimed that "existing threat models assume test-time samples will be labeled", and TePA (Cong et al., 2024) relies on access to the entire source dataset to train a substitute model. Actually, TePA also is a label-free attack, and it does not need access to the entire source dataset. It only needs a surrogate dataset that has a similar distribution of the source dataset. I suggest that the authors reorganize the introduction of related work.
>
>
>
> Thank you for pointing this out. We have included this discussion in the Introduction section of the updated manuscript.
>
>
>
> > How many batches does FCA need to effectively decrease the target model’s performance?}}
>
>
>
> We treat each batch as an independent trial and calculate error rates solely on unperturbed benign samples. Table 1 reports error rates for unperturbed benign samples while 20\% of the batch samples are adversarially perturbed. After each adaptation step, the victim TTA state is reset to its pre-adaptation state, as if no adversarial samples were included. Thus, FCA effectively decreases the target model's performance with just a single batch.
>
> > In experiments like Table 1, this paper does not compare the attack success rate with TePA (Cong et al., 2024).}}
>
> We have added additional results on Table 1 that compares how black-box methods fares against FCA in our evaluation settings. We did not include experiments of Imagenet-C for TePA, as for this dataset no surrogate black box model was provided by the authors.
>
> References:
>
> [1] Tent: Fully test-time adaptation by entropy minimization, ICLR 2021.
>
> [2] Rdumb: A simple approach that questions our progress in continual test-time adaptation, NIPS 2024

---

> > ### Author Response · Authors · 2024-11-24
> >
> > > The proposed attack seems not to be stealthy. In other words, FCA needs a high poisoning rate (Table 3).
> >
> >
> >
> > Although in the initial experimental setup description we have mentioned that the batch size of 200 will be considered in all experiments if not otherwise specified, Table 3 should include the percentage of poisoned samples for clarity . The FCA attack is similarly  and more stealthy compared to existing work  DIA \citep{wu2023uncovering} and  TePA \citep{cong2024test} respectively. Even with 10 malicious samples (5\% of the whole batch), FCA can increase the error rate more than 4\% for unperturbed benign samples in CIFAR 10C compared to TePA.
> >
> >
> >
> > > In (Cong et al., 2024). they also proposed several defenses, which are not discussed in this paper.}}
> >
> >
> >
> > We have investigated the existing defense mechanisms mitigating adversarial attack on TTA [1,2]. Moreover, we have included experiments involving adversarial training on the source DNN as a defense mechanism. A desirable property of a defense mechanism against our FCA or similar threats is that it should not degrade the performance of the inference involving TTA. Since the other presented defense mechanism in [3] degrades the performance to a significant extent, we have not included them as any potential defense mechanism. For example, involving JPEG compression as defense degrades the normal inference accuracy by approximately (10-50\%) for fog corruption [3].
> >
> >
> >
> > > This paper considers five target TTA methods, which all need the test samples to be fed in a batch-by-batch manner. However, some TTA methods like TTT [R1] can accept only one sample to adjust the model. I am curious if FCA can attack such kinds of TTA methods.
> >
> > Thank you for pointing this out. Please see the common response to the reviewers.
> >
> > References:
> >
> > [1] SoTTA: Robust Test-Time Adaptation on Noisy Data Streams, NIPS 2024
> > [2] Medbn: Robust test-time adaptation against malicious test sample, CVPR 2024.
> > [3] Test-time poisoning attacks against testtime adaptation model, IEEE SP 2024.

---

### Official Review · Reviewer_k7PY · 2024-11-02

**Soundness:** 3
**Presentation:** 3
**Contribution:** 3
**Rating:** 6
**Confidence:** 3

**Summary:**

The authors study the robustness of test-time adaptation (TTA) methods and propose a novel label-free attack. They argue that the assumption of the attacker having true labels on the test set is too strong for the real-life scenarios and without it the existing DIA attack is much less effective. However, the FCA attack proposed in the paper, that uses the idea of neural collapse, remains effective in the label-free scenario which emphasizes the need to further study the TTA robustness.

**Strengths:**

**Originality**: label-free attacks on test-time adaptation seem to be a novel direction. The paper demonstrates the weaknesses of existing defence mechanisms (Section 7.1)

**Quality**: the results are supported with numerous experiments on different datasets (CIFAR10-Cm CIFAR100-C, ImageNet-C) and have theoretical justification (Theorem 1 with proof in Appendix). Ablation study with respect to different loss components is performed (Table 4).

**Clarity**: the paper is in general well structured.

**Significance**: robustness of test-time adaptation is important in real life applications.

**Weaknesses:**

**Quality:**

1)	The experiments are performed only with the ResNet architecture variants (line 316). Empirical results would benefit from including some other popular architecture e. g. one based on a vision transformer. At least in the Appendix.

**Clarity:**

1)	Figure 1a: the process in the Figure 1a is hard to grasp for an outside reader. The diagram flow is confusing. Consider making it clearer where the flow starts and ends.  The overview figure has to be more intuitive to help the reader better understand the paper idea and not to confuse them. Consider simplifying it or breaking into several Figures. The top dotted line is missing an arrow on its end.

2)	Please elaborate on how the pseudo labels are defined or add a citation (line 364)

**Typos and minor comments:**

1)	Line 090 - “as” not needed: This makes our proposed attack **as** a prevalent threat that requires countermeasure

2)	Line 134 - additional “the”: and **the** the set

3)	Line 172 - Why is “Nearest Neighbour Classifier” abbreviated as NCC and not NNC?

4)	Line 265 - the sentence is too complex and poorly constructed, it is hard to understand its point. The more the last layer activations collapse to class mean vector, the likelihood of having a region close to it where the actual prediction deviates if the sample is projected into that region decreases

5)	Line 314 - delete one “use”: we use the use the provided data.

6)	It would be good to include “w/o Attack” value (Table 1) in the Table 4 to better the relative improvement of each variant. The L_nc + L_col would also be good to include for completeness. Perhaps there is some synergy between these two terms that make their combination stronger than the sum of each term.

**Questions:**

1) Why is the perturbation initialized with 0.5 and not e. g. as random noise? (Algorithm 1, step 2) Wouldn’t the optimization benefit from this?

---

> ### Author Response · Authors · 2024-11-24
>
> > The experiments are performed only with the ResNet architecture variants (line 316). Empirical results would benefit from including some other popular architecture e. g. one based on a vision transformer. At least in the Appendix.
>
>
>
> Thank you for pointing this out. We have not included any Vision Transformer models in our benchmarking because the compared TTA algorithms cannot be directly applied to Vision Transformers due to the absence of batch normalization layers. The original benchmark TTA methods also did not experiment with Vision Transformer models. We attempted to adapt the layer normalization layers in Vision Transformers instead of \gls{bn} layers while keeping other aspects unchanged. However, even with a very small learning rate, adapting the parameters of layer normalization resulted in performance degradation rather than improvement. This phenomenon was observed across all three benchmark datasets and five TTA methods. Therefore, without further exploration, we cannot yet comment on the effectiveness of FCA against Vision Transformers.
>
> We have studied the effectiveness of FCA in MobileNet architecture family and have shown the results in Table 7 of the revised manuscript. Our findings show that TTA methods exhibit similar vulnerabilities on CIFAR-10C and CIFAR-100C datasets as those observed with ResNet variants. However, for the ImageNet-C benchmark, MobileNet-v2 proved to be even more vulnerable, with performance degradation under FCA being  $\sim$ 10\% greater compared to the ResNet-50.
>
>  > Please elaborate on how the pseudo labels are defined or add a citation (line 364)
>
> Pseudo labels can be obtained using the current model's prediction probabilities (soft PL) or the argmax of these predictions (hard PL) for benchmarking. Both methods have similarly low efficacy on attack strength. For our experimental evaluations, we use hard pseudo labels and refer to them simply as PL for brevity.
>
> > About Typos and minor comments:
>
> We appreciate the reviewer’s meticulous reading of our paper and the valuable suggestions. In the updated manuscript, we have incorporated all the recommendations accordingly.
>
> > Figure 1a: the process in the Figure 1a is hard to grasp for an outside reader. The diagram flow is confusing. Consider making it clearer where the flow starts and ends. The overview figure has to be more intuitive to help the reader better understand the paper idea and not to confuse them. Consider simplifying it or breaking into several Figures. The top dotted line is missing an arrow on its end.
>
> We apologize for the previously unclear overview figure. In the revised version, we have distinctly labeled the different systems and clarified the information flow. We hope this updated figure is easier to understand.
>
> >  Why is the perturbation initialized with 0.5 and not e. g. as random noise? (Algorithm 1, step 2) Wouldn’t the optimization benefit from this?
>
>
>
> Thank you for pointing this out. In Algorithm 1, step 2, we initialized the perturbation with a value of 0.5 instead of random noise to align with the hyperparameters of the benchmark attack [1] and defense mechanism [2]. This approach ensures a direct and fair performance evaluation under equivalent conditions. However, you correctly noted that initializing with random noise could theoretically explore a broader space and potentially identify stronger adversarial perturbations. We appreciate this feedback and have run additional experiments to understand the impact of initialization. The findings of this experiment align with the suggestion, consistently creating stronger perturbations (0.04 - 0.1\% error rate increase) and proving to be a superior initialization choice.
>
> ### (% Error) Comparison with Different Initialization Schemes on CIFAR-100C Dataset
>
> | **Initialization Scheme** | **TeBN** | **TENT** | **EATA** | **SAR**  | **SoTTA** |
> |----------------------------|----------|----------|----------|----------|-----------|
> | Const (0.5)               | 38.87    | 37.95    | 40.15    | 37.33    | 38.37     |
> | Random                    | 38.91    | 37.98    | 40.27    | 37.28    | 38.43     |
>
>
>
>
>
> References:
>
> [1] Uncovering adversarial risks of test-time adaptation, ICML 2023.
>
> [2] Medbn: Robust test-time adaptation against malicious test sample, CVPR 2024.

---

> > ### Comment · Reviewer_k7PY · 2024-11-27
> > **Response**
> >
> > Thank you for addressing the issues raised in my review. After making myself familiar with the other reviews and responses I decided to keep my original score.

---

### Official Review · Reviewer_JDoU · 2024-11-03

**Soundness:** 3
**Presentation:** 2
**Contribution:** 3
**Rating:** 6
**Confidence:** 3

**Summary:**

The paper concentrates on the adversarial vulnerability in test-time adaptation (TTA). To explore more practical adversarial attacks, the paper designs a novel attack algorithm, named Feature Collapse Attack (FCA), which can achieve comparable performance without access to labeled test samples. This method consists of three loss designs, which are Nearest Centroid Loss, Feature Collapse Loss and Feature Scattering Loss. Experimental results demonstrate the outstanding performance of the proposed method.

**Strengths:**

+ **New Scenario**: The paper explores a novel scenario, where the attacker has no access to the true labels.
+ **Simple Implementation**: The proposed method is relatively easy to implement and follow.

**Weaknesses:**

+ The number of baseline attack methods is limited. More comparison with multiple attacks is expected.
+ In line 134, there is a repeated word "the".

**Questions:**

The paper considers the worst-case scenario where the attacker has knowledge of the TTA algorithm and has white-box access to the latest updated model parameters provided by some malicious insider. Under this assumption, is it reasonable to hypothesize that the attacker has no access to the true labels?

---

> ### Author Response · Authors · 2024-11-24
>
> > The number of baseline attack methods is limited. More comparison with multiple attacks is expected
>
> Thank you for pointing this out. Understanding the security vulnerabilities of TTA is in its early stage of scrutiny, as such, the number of existing methods to compare with is limited.  As pointed out by Reviewer AcHd, we have included an additional baseline TePA [1] in the revised manuscript. Should you have some specific additional baselines in mind, we would be glad to include them in the updated manuscript when the results are available.
>
>
>
> > The paper considers the worst-case scenario where the attacker has knowledge of the TTA algorithm and has white-box access to the latest updated model parameters provided by some malicious insider. Under this assumption, is it reasonable to hypothesize that the attacker has no access to the true labels?
>
> We thank the reviewer for raising this point. Although we consider white-box access to evaluate the worst-case scenario, we assume the adversary does not have access to true labels. The rationale behind this choice is that TTA updates a DNN on unlabeled test data available in an online manner before performing inference. If true labels were available, the entire TTA setup would become redundant, allowing us to concentrate on the challenges of efficient fine-tuning. The best an adversary can do is label the data using the latest updated DNN, which we term as pseudo labels. Interestingly, even when these models are moderately accurate (with CIFAR-10C accuracy above 80\%), the existing attacks proposed by [2] become almost ineffective.
>
> References:
>
> [1] Test-time poisoning attacks against test time adaptation models, IEEE SP 2024.
>
> [2] Uncovering adversarial risks of test-time adaptation, ICML 2023.

---

> > ### Comment · Reviewer_JDoU · 2024-12-03
> >
> > Thank you for addressing my questions. I decided to slightly raise my score.

---

### Official Review · Reviewer_fWBZ · 2024-11-04

**Soundness:** 2
**Presentation:** 2
**Contribution:** 2
**Rating:** 6
**Confidence:** 2

**Summary:**

This paper proposes a novel label-free adversarial attack against Test-Time Adaptation (TTA) algorithms, called Feature Collapse Attack (FCA). The authors argue that existing TTA attack methods unrealistically assume access to test data labels, making them impractical for real-world scenarios. They address this gap by formulating a new attack that leverages the principles of Neural Collapse (NC) to degrade TTA performance without needing ground truth labels.

**Strengths:**

1. The paper addresses a significant limitation of existing TTA attack research by introducing a label-free approach. This is crucial for practical security evaluations of TTA, as real-world adversaries are unlikely to have access to test labels.
2. The attack design is well-motivated by the theory of Neural Collapse, providing a clear rationale for the chosen loss functions.
3. FCA achieves competitive performance against state-of-the-art attacks that utilize labels, and even surpasses them in some cases. This demonstrates its effectiveness and potential impact.
4. The authors evaluate their attack across multiple datasets and TTA methods, providing a thorough assessment of its generalizability.

**Weaknesses:**

See Questions

**Questions:**

1. **Assumption of White-Box Access:** The attack assumes white-box access to the model parameters, which might not always be realistic. Analyzing the attack's performance under more restricted settings, such as black-box or gray-box access, would provide a more complete picture of its capabilities.
2. **Attack Algorithm:** The attack part of the FCA algorithm is almost entirely borrowed from the PGD in the adversarial sample attack, which will affect the contribution of this paper. Furthermore, does this suggest that the method could be well enhanced by using more powerful attacks?
3. **The Robustness of Victim Model:** The effectiveness of the proposed attack may be limited when targeting inherently robust models **[R]** or those with defense mechanisms (adversarial training), since it builds upon adversarial attack principles. How significantly does the attack's performance degrade when faced with robust or defended victim models?

**[R]** MEMO: Test Time Robustness via Adaptation and Augmentation.

---

> ### Author Response · Authors · 2024-11-24
>
> > Assumption of White-Box Access: The attack assumes white-box access to the model parameters, which might not always be realistic. Analyzing the attack's performance under more restricted settings, such as black-box or gray-box access, would provide a more complete picture of its capabilities.
>
> Thank you for pointing this out. Please see the common response to the reviewers.
>
> > Attack Algorithm: The attack part of the FCA algorithm is almost entirely borrowed from the PGD in the adversarial sample attack, which will affect the contribution of this paper. Furthermore, does this suggest that the method could be well enhanced by using more powerful attacks?
>
>
>
> Thank you for your remark. The main contribution of this paper is understanding how much the existing attack mechanism [1] is dependent on the availability of the true labels of test data available in an online manner in TTA settings, as well as the design of an attack mechanism based on a better understanding of the adversarial risk of TTA. We believe that using PGD as the optimization process does not weaken any of our claims or findings. PGD has been widely used to understand the security and robustness of a wide variety of learning methods and systems, including contrastive learning [2], TTA [1] and adversarial training [3]. Nevertheless, we acknowledge that the proposed method could be enhanced by using more powerful attacks or variants of PGD like having adaptive step-size and perturbation budget and we thank the reviewer for raising this point, which we plan to explore in future work.
>
>
>
> > The Robustness of Victim Model: The effectiveness of the proposed attack may be limited when targeting inherently robust models [R] or those with defense mechanisms (adversarial training), since it builds upon adversarial attack principles. How significantly does the attack's performance degrade when faced with robust or defended victim models?
>
> Thank you for this remark. Please refer to the common response about benchmarking the effectiveness of FCA on [4].
>
>
>
> To compare the effectiveness of TTA against adversarially-trained models, we have conducted additional experiments using models robust to both domain shift and adversarial perturbation. The results are presented in Tables 8 and 9 in the Appendix, respectively. We notice that adversarially-trained DNNs are highly effective against TTA when the same perturbation constraint$(\epsilon_{∞}=8/255)$ is used for both crafting adversarial examples and training the source \gls{dnn}. However, with a different perturbation budget, such as an $l_{2}$ norm constraint of $(\epsilon_{2}=0.5)$, TTA can still degrade performance by approximately 4\%. Furthermore, for the CIFAR-100C dataset, adversarially-trained source DNNs result in more than 10\% increase in error rate during adaptation with non-malicious data. This is unexpected, as TTA  is generally intended to handle online data batches without adversarial perturbation, raising concerns about the robustness-utility trade-off in deploying adversarially trained DNNs.
>
>
> ### Table: (% Error) Comparison on Adversarially Trained Models
>
> | **Dataset**       | **Evaluation Setup**        | **TeBN** | **TENT** | **EATA** | **SAR** | **SoTTA** |
> |--------------------|-----------------------------|----------|----------|----------|----------|-----------|
> | **CIFAR10-C**      | Unattacked (Standard)      | 17.14    | 16.98    | 19.21    | 16.88    | 16.42     |
> |                    | Unattacked (Adv trained)   | 19.21    | 16.22    | 18.44    | 17.91    | 15.40     |
> |                    | FCA (ε∞ = 8/255)           | 21.44    | 18.01    | 20.25    | 19.83    | 17.17     |
> |                    | FCA (ε₂ = 0.5)             | 23.45    | 20.14    | 22.03    | 21.55    | 19.03     |
> | **CIFAR100-C**     | Unattacked (Standard)      | 31.27    | 30.91    | 31.87    | 30.90    | 29.30     |
> |                    | Unattacked (Adv trained)   | 42.04    | 41.59    | 42.14    | 41.51    | 41.04     |
> |                    | FCA (ε∞ = 8/255)           | 43.01    | 42.57    | 20.25    | 42.79    | 41.85     |
> |                    | FCA (ε₂ = 0.5)             | 46.44    | 45.22    | 44.55    | 45.01    | 44.76     |
>
> Additionally, adversarial training is known to reduce accuracy on clean data [5,6]. Thus, further analysis is required to develop test-time defenses that are both computationally efficient yet effective against TTA without impairing TTA performance on clean or benign samples from different domains.

---

> ### Author Response · Authors · 2024-11-24
>
> To further understand how the robustness of the source DNN influences FCA, we have analyzed the performance of TTA against source DNNs known for their robustness to distribution shifts. Specifically, we have utilized the WideResNet-28 model trained with AugMix [7] from Robustbench [8]. While AugMix-trained models are effective in enhancing robustness against various distribution shifts, they remain vulnerable to attacks when deployed for TTA.
>
> ### (% Error) Comparison on Robust Models
>
> | **Dataset**   | **Evaluation Setup**   | **TeBN** | **TENT** | **EATA** | **SAR**  | **SoTTA** |
> |---------------|-------------------------|----------|----------|----------|----------|-----------|
> | **CIFAR10-C** | Unattacked (AugMix)    | 15.37    | 14.81    | 16.47    | 14.53    | 14.11     |
> |               | FCA                    | 24.33    | 23.21    | 24.98    | 23.05    | 22.87     |
> | **CIFAR100-C**| Unattacked (AugMix)    | 29.34   | 28.77    | 30.21    | 28.55    | 27.87     |
> |               | FCA                    | 36.41    | 35.22    | 37.02    | 34.77   | 34.28     |
>
>
>
>
>
>
>
>
>
>
>
>
>
>
>
>
> References:
>
> [1] Uncovering adversarial risks of test-time adaptation, ICML 2023.
>
> [2] Indiscriminate poisoning attacks on unsupervised contrastive learning, ICLR 2022.
>
> [3] On the vulnerability of adversarially trained models against two-faced attacks, ICLR 2024.
>
> [4] MEMO: Test Time Robustness via Adaptation and Augmentation, NIPS 2022.
>
> [5] Theoretically principled trade-off between robustness and accuracy, ICML 2019.
>
> [6] Robustness may be at odds with accuracy, ICLR 2019.
>
> [7] Augmix: A simple data processing method to improve robustness and uncertainty, ICLR 2020.
>
> [8] Robustbench: a standardized adversarial robustness benchmark, NIPS 2021.

---

### Author Response · Authors · 2024-11-24
**Common Response to all Reviewers**

**Choice of White-box Attack Setting**

Although we acknowledge that black-box attacks against TTA methods is more practical, in this paper we are aiming at understanding the fundamental robustness of  existing TTA methods, which can only be investigated by assuming the worst-case scenario.  We remark that understanding vulnerabilities in white-box settings will facilitate the creation of defense strategies that are also effective in black-box settings. We point out that our assumption of white-box setting perfectly aligns with prior work [1] on understanding the fundamental vulnerabilities of TTA as well as  possible defenses [2].



Besides achieving a fundamental understanding of existing vulnerabilities, white-box attacks are extremely relevant in scenarios with malicious insiders, which are common in settings where the AI/ML algorithms are created locally and then shared with other users. For example, in the case of cellular networks using the Open Radio Access Network (O-RAN) paradigm [3], AI/ML models are trained by local O-RAN providers and then shared in common application marketplaces [4,5]. In such case, it is extremely likely that a malicious O-RAN provider will have perfect knowledge not only of the AI/ML model but also of its updated weights after TTA, for example, to perform additional fine-tuning with local data [12].


Finally, we have conducted additional experiments in  grey-box settings, i.e., where the attacker knows the model architecture and initial weights. We have presented these results in Table 6. Our proposed attack mechanism remains highly effective in this setting as well, reducing attack effectiveness by about 2\% only across different TTA methods.


**Regarding exclusion of single sample TTA methods**



We have not considered single-sample TTA methods such as [9,10] since they require fundamentally different considerations in terms of threat model and attack design. In stark opposition, our work studies how unperturbed, non-malicious samples are influenced by the presence of adversarially perturbed samples when true labels are unavailable.



In single-sample scenarios, an adversary would need to craft attacks tailored to each test sample while accounting for augmentations, which aligns more closely with evasion attacks [6,7]. Conversely, our adversarial model is closely related to TTA poisoning [8]. The former case demands a specialized approach and attack mechanism, which deserves a separate investigation. We also point out that single-sample methods often incur substantial computational costs (e.g., multiplying the number of forward and backward passes by the batch size) without significantly improving robustness against distribution shifts [11], which limits their applicability.

References:

[1] Uncovering adversarial risks of test-time adaptation, ICML 2023

[2] Medbn: Robust test-time adaptation against malicious test sample, CVPR 2024.

[3] Understanding O-RAN: Architecture, Interfaces, Algorithms, Security, and Research Challenges. arXiv preprint arXiv:2202.01032, 2022.

[4] Managing o-ran networks: xapp development from zero to hero. arXiv preprint arXiv:2407.09619, 2024.

[5] Intelligence and Learning in O-RAN for Data-driven NextG Cellular Networks. IEEE Communications Magazine, October 2021

[6] Explaining and harnessing adversarial examples, arXiv preprint arXiv:1412.6572, 2014.

[7] Towards evaluating the robustness of neural network, IEEE SP 2017.

[8] Test-time poisoning attacks against testtime adaptation model, IEEE SP 2024.

[9] Test-time training with self-supervision for generalization under distribution shift, ICML 2020

[10] Memo: Test time robustness via adaptation and augmentation, NIPS 2022.

[11] Efficient test-time model adaptation without forgetting, ICML 2022

[12]  Open-RAN Gym: An Open Toolbox for Data Collection and Experimentation with AI in O-RAN, WCNC 2022

---

### Comment · Area_Chair_Ap58 · 2024-12-01

Dear Reviewers,

The authors have provided responses - do have a look and engage with them in a discussion to clarify any remaining issues as the discussion period is coming to a close in less than a day (2nd Dec AoE for reviewer responses).

Thanks for your service to ICLR 2025.

Best,
AC

---

### Meta-Review · Area_Chair_Ap58 · 2024-12-24

**Metareview:**

This paper develops a new adversarial attack on test-time adaptation (TTA) algorithms that does not require access to ground truth labels at test-time. The attack is motivated by previous work on neural collapse and is shown to defeat current defense mechanisms in extensive evaluations on multiple datasets.

Key strengths of the paper are that it addresses the robustness of an emerging class of methods with a novel approach that is well-motivated by theoretical arguments. Extensive experimental results in white-box and more realistic grey-box attack scenarios show that the attack is effective. On the other hand, the paper does not cover the class of single-sample test-time adaptation methods, and does not include a comprehensive evaluation on the axis of model architectures.

Overall the paper presents a novel, well-motivated attack for TTA that is effective in a reasonable grey-box setting (for which there are few existing attacks) and the AC recommends acceptance. The work will spur the development of improved defenses in the future. The AC also suggests that the grey-box attack results be included in the main text of the final version of the paper.

**Additional Comments On Reviewer Discussion:**

There were common concerns about the white-box threat model used in the initial version of the paper, as well as a lack of comparison to relevant baselines and other model architectures. The authors provided additional experiments on the grey-box setting, and included comparisons with an additional baseline (TePA) as well as results on a different model architecture (MobileNet). The AC believes these additional results address the initial concerns of the reviewers.

There was also a concern that the proposed attack does not cover the full class of TTA methods, specifically those that take a single-sample at adaptation time. The authors provided a reasonable explanation that should probably be also included in the final version of the paper.

Finally there was a concern that robust initial models would defend against the proposed attack; the authors included additional results that showed that such models were still vulnerable, thus addressing this concern.

Overall the addition of grey-box experiments was key to convincing the AC towards acceptance, as it addresses the key concern of multiple reviewers of a reasonable threat model in the TTA setting.

---

### Decision · Program_Chairs · 2025-01-22

Accept (Poster)